# Single-Loop Penalty Methods for Bilevel Reinforcement Learning

## Abstract

Bilevel reinforcement learning (RL) models a leader that optimizes an outer objective while the follower solves an inner policy optimization problem. Penalty reformulations turn this constrained problem into a single-level surrogate whose minimizers approximate bilevel solutions, and recent work gave principled penalties with closed-form gradients and first-order convergence. Yet existing algorithms are double-loop: each outer step calls an inner best-response oracle, yielding extra logarithmic overhead. We present *PBRL-SL*, a *single-loop* penalty method that dispenses with the inner oracle. A tracking policy follows the follower's optimal response with one mirror-descent/policy-gradient step; a Lyapunov argument absorbs the resulting gradient bias. Under standard regularity, PBRL-SL achieves $\tilde{O}(\lambda\varepsilon^{-2})$ projected-gradient stationarity, matching prior iteration order while being simpler to implement.

## 1 Introduction

Bilevel optimization ties two decisions together: the outer variable $x$ controls an environment or a learner, while the inner variable $y$ solves a problem induced by $x$. In bilevel RL the inner problem is not a benign convex model; it is policy optimization in an MDP or Markov game. This setting covers reward shaping, incentive design, and RL from human feedback (RLHF), where the outer decision shapes rewards, dynamics, or data collection, and the follower returns a policy that is optimal for the shaped problem. In these applications the lower objective—the discounted return—is non-convex in the policy, so classical implicit-gradient methods relying on strong convexity or uniform PL conditions are inapplicable.

Penalty reformulations have recently emerged as a robust path forward. Two penalties are especially effective. The *value penalty* measures how far a candidate policy is from the inner optimum in terms of regularized value. The *Bellman penalty* measures how far the policy is from minimizing a strongly convex surrogate built from optimal $Q$-values; with a positive regularization parameter $\tau$, the follower's optimal policy becomes unique and the penalty is zero exactly at optimality. For both penalties, closed-form gradients with respect to the outer parameters can be written, and the penalized objective is smooth under standard modeling assumptions. These ingredients enable projected first-order algorithms with finite-time guarantees. We will reuse these facts and cite them precisely when they are invoked.

Despite this progress, there is a practical bottleneck. Current penalty-based methods update $(x, y)$ only after obtaining an *approximately optimal* inner policy for the current $x$. This inner policy is produced by running a policy mirror-descent or policy-gradient routine to near-convergence; the overall complexity therefore carries a logarithmic overhead from the inner loop, and in practice the inner loop often dominates wall-clock time. This burden is explicit in the summary of convergence results in the prior work and in their algorithmic template, which requires an inner best-response oracle at each outer step.

Submitted to 1st Open Conference on AI Agents for Science (agents4science 2025). Do not distribute.

This paper asks a direct question: can we keep the same penalty framework and convergence order but *remove* the inner loop? Our answer is yes. We introduce a *single-loop* algorithm, PBRL-SL, that maintains a tracking policy meant to shadow the follower's optimal response. Each iteration performs one light-weight tracking step and one projected outer step that uses a biased penalty-gradient estimator. The analysis shows that the tracking error contracts up to a drift term caused by changes in $x$, and that the gradient bias is controlled by this error and the outer variation. A Lyapunov function combines the penalized objective and the squared tracking error and yields a descent inequality. Choosing stepsizes to balance contraction and drift leads to the same $\tilde{O}(\lambda \varepsilon^{-2})$ stationarity guarantee as in the double-loop method, now *without* calling any inner oracle.

Beyond the theorem, the single-loop structure matters in practice. In RLHF pipelines, where reward modeling and policy optimization interleave, avoiding an inner best-response substantially simplifies engineering and reduces end-to-end latency. In incentive design and Stackelberg control, where environment calls are costly, replacing inner convergence by a single policy step reduces samples per outer iteration. We revisit these scenarios in a dedicated discussion section.

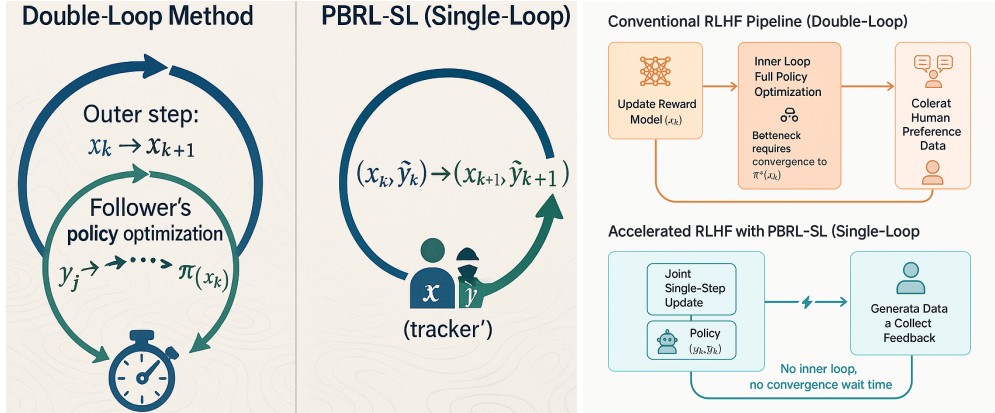

Figure 1: Comparison of double-loop and single-loop methods for bilevel reinforcement learning (left) and their specific application to reinforcement learning from human feedback (RLHF) (right).

## 2 Related Works

This work sits at the intersection of bilevel optimisation, reinforcement learning (RL) and alignment from human feedback. We briefly review the literature on each thread and emphasise the differences from the proposed single-loop penalty method.

**Penalty-based bilevel RL.** Classical bilevel optimisation methods treat the outer variable and the inner variable asymmetrically. (5) introduced a first-order penalty framework for supervised bilevel problems under error-bound or Polyak–Łojasiewicz conditions and proved convergence at order $\tilde{O}(\lambda \epsilon^{-1})$ while relying on an inner-loop oracle to solve the lower-level problem. Our method builds on their penalty philosophy but removes the inner oracle via tracking and Lyapunov absorption. Recently, (8) developed a principled penalty-based framework specifically for bilevel RL and reinforcement learning from human feedback (RLHF). They introduce value and Bellman penalties to measure the deviation of a candidate policy from the inner optimum; closed-form penalty gradients are derived, the penalised objective is smooth, and first-order convergence guarantees are established. However, their algorithms remain double-loop: each outer step calls a best-response oracle for the lower policy. By contrast, our PBRL-SL algorithm follows only one policy-gradient/mirror-descent step per outer iteration and uses a Lyapunov argument to control the resulting gradient bias.

**Hyper-gradient and single-loop bilevel RL.** Another line of work characterises bilevel RL through the hyper-gradient. (9) develop a fully first-order hyper-gradient for bilevel RL without assuming lower-level convexity by exploiting fixed-point equations for regularised RL. They propose model-based and model-free algorithms with convergence rate $O(\epsilon^{-1})$ and introduce a stochastic

variant with iteration and sample complexity guarantees (9). (10) frame contextual bilevel RL (CB-RL) as a Stackelberg game where a leader and random context jointly determine a contextual MDP; they develop a stochastic hyper-policy-gradient descent (HPGD) algorithm that estimates hyper-gradients from followers' trajectories. Our work differs in that we stick to penalty formulations and derive a simple tracking rule, avoiding explicit hyper-gradient estimation and performing only one follower update per outer step. Within bilevel optimisation more broadly, (7) proposed a fully single-loop algorithm (FSLA) for supervised bilevel problems by approximating the hyper-gradient and maintaining a state variable; they prove $O(\epsilon^{-2})$ convergence without Hessian inversion. While sharing the single-loop spirit, FSLA assumes strongly convex inner problems and cannot handle RL followers. (16) recently proposed an efficient curvature-aware hyper-gradient approximation that incorporates curvature information into implicit gradient estimation and improves computational complexity. Their method targets general bilevel problems and is complementary to our penalty-based RL approach.

**Bilevel RL for alignment and incentives.** Bilevel RL has become a natural framework for formalising incentive design and policy alignment tasks. Chakraborty *et al.* propose PARL, a unified bilevel framework for policy alignment in RLHF (11). They explicitly parameterise the distribution of the alignment objective (reward design) by the lower-level optimal policy, turning RLHF into a stochastic bilevel problem, and develop A-PARL with $O(1/T)$ sample complexity. Thoma *et al.* (CB-RL) allow exogenous context and multiple followers and design HPGD with hyper-gradient estimates (10). Makar-Limanov *et al.* model RLHF as a Stackelberg game between a language model and a preference model; they devise a nested gradient descent–ascent algorithm to approximate the Stackelberg equilibrium and show empirically that the resulting language model outperforms other RLHF methods (12). Li *et al.* argue that the standard three-stage RLHF pipeline wastes data and propose Alignment with Integrated Human Feedback (AIHF) to jointly learn the reward and policy models using both demonstration and preference data; their algorithm enjoys finite-time performance guarantees and significantly outperforms existing alignment baselines (13). These works highlight that bilevel structure and sequential decision making are central to alignment; our single-loop method contributes by offering a more efficient optimisation routine for such bilevel RL problems.

**Reinforcement learning from human preferences.** Our outer objective and examples draw on the literature of RLHF. Christiano *et al.* introduced learning from human preferences, where a reward model is trained from pairwise comparisons of trajectory segments and used to train an RL agent; they demonstrated that complex behaviours can be learned without a reward function and that only a small amount of human feedback is required (17). In contrast, subsequent alignment works, including PARL and AIHF, frame RLHF as a bilevel optimisation problem. Our penalty formulation fits naturally into this framework by viewing reward design as the outer variable and policy training as the inner variable.

## 3 Methodology

We first fix notation and restate, self-contained, the penalty ingredients we will use. All facts in this subsection summarize established results and are stated without referring to original numbering; citations appear inline.

### 3.1 Preliminaries and Notation

Let $\mathcal{M}_\tau(x) = (\mathcal{S}, \mathcal{A}, r_x, P_x, \tau h)$ be a finite MDP parameterized by $x \in X \subset \mathbb{R}^{d_x}$, with discount $\gamma \in [0, 1)$ and a statewise 1-strongly convex regularizer $h = (h_s)_s$ applied with weight $\tau \geq 0$. Policies $\pi$ are in a convex class $\Pi$ (tabular or softmax). For a policy $\pi$,

$$V^\pi_{\mathcal{M}_\tau(x)}(s) = \mathbb{E}\Big[ \sum_{t \geq 0} \gamma^t \big( r_x(s_t, a_t) - \tau h_{s_t}(\pi(\cdot|s_t)) \big) \mid s_0 = s, \pi \Big], \tag{1}$$

$$Q^\pi_{\mathcal{M}_\tau(x)}(s, a) = r_x(s, a) + \gamma \, \mathbb{E}_{s'} V^\pi_{\mathcal{M}_\tau(x)}(s'). \tag{2}$$

Given a full-support distribution $\rho$, write $V^\pi_{\mathcal{M}_\tau(x)}(\rho) = \mathbb{E}_{s \sim \rho} V^\pi_{\mathcal{M}_\tau(x)}(s)$. The follower solves $\max_{\pi \in \Pi} V^\pi_{\mathcal{M}_\tau(x)}(\rho)$; denote the (possibly unique) optimal policy by $\pi^\star(x)$.

118 The bilevel RL problem is

$$\min_{x \in X, \, y \in Y} f(x,y) \text{ s.t. } \pi_y \in \arg\max_{\pi \in \Pi} V^{\pi}_{\mathcal{M}_\tau(x)}(\rho),$$

119 where $y$ parametrizes $\pi_y$ and $f$ is smooth.

## 3.2 Penalty functions: self-contained recap

121 **Value penalty.** Define

$$p_{\text{val}}(x,y) := \max_{\pi \in \Pi} V^{\pi}_{\mathcal{M}_\tau(x)}(\rho) - V^{\pi_y}_{\mathcal{M}_\tau(x)}(\rho).$$

122 Then $p_{\text{val}}(x,y) \geq 0$, and $p_{\text{val}}(x,y) = 0$ iff $\pi_y$ is optimal for the inner MDP. Under mild regularity
123 ensuring gradients of $V^{\pi}_{\mathcal{M}_\tau(x)}$ with respect to $x$ agree across optimal policies, $x \mapsto p_{\text{val}}(x,y)$ is
124 differentiable with

$$\nabla_x p_{\text{val}}(x,y) = -\nabla_x V^{\pi_y}_{\mathcal{M}_\tau(x)}(\rho) + \nabla_x V^{\pi^\star(x)}_{\mathcal{M}_\tau(x)}(\rho), \quad \nabla_y p_{\text{val}}(x,y) = -\nabla_y V^{\pi_y}_{\mathcal{M}_\tau(x)}(\rho).$$

125 A gradient-dominance inequality holds on convex $\Pi$, connecting the value gap to a linearized ascent
126 residual. These facts are standard in regularized policy optimization and are established in the
127 penalty-based bilevel RL literature.

128 **Bellman penalty.** Let $q_s(x) \in \mathbb{R}^{|\mathcal{A}|}$ collect $-\max_{\pi \in \Pi} Q^{\pi}_{\mathcal{M}_\tau(x)}(s,a)$ over $a$. Define

$$g(x,y) = \mathbb{E}_{s \sim \rho}\big[\langle y_s, q_s(x)\rangle + \tau h_s(y_s)\big], \quad v(x) = \min_{y \in Y} g(x,y), \quad p_{\text{bel}}(x,y) = g(x,y) - v(x).$$

129 Because $h_s$ is 1-strongly convex, $g(x,\cdot)$ is $\tau$-strongly convex, so $p_{\text{bel}} \geq 0$. For any $\tau > 0$, the
130 follower's optimal policy is unique, and $p_{\text{bel}}(x,y) = 0$ exactly at this policy. Moreover, under
131 continuity of $\nabla_x Q^\pi$ and an irreducibility condition that guarantees stable visitation distributions,
132 both $\nabla_x g$ and $\nabla_x v$ admit closed forms in terms of $\nabla_x Q^\pi$, hence $\nabla_x p_{\text{bel}}$ is explicit; with smooth
133 parameterizations, $p_{\text{bel}}$ is Lipschitz-smooth. *In implementation we will substitute the unknown $q(x)$*
134 *by a plug-in term built from the tracker (defined below); the bias induced by this substitution is*
135 *controlled in Lemma 2.*

136 **Penalized objective.** For $p \in \{p_{\text{val}}, p_{\text{bel}}\}$, define $F_\lambda(x,y) = f(x,y) + \lambda p(x,y)$. Local minimizers
137 of $F_\lambda$ approximate feasible solutions of the bilevel problem when $\lambda$ exceeds a data-dependent
138 threshold; $F_\lambda$ is smooth under the above conditions. These landscape and smoothness statements
139 were developed for value/Bellman penalties and will be used as black boxes below.

140 **Existing facts used as black boxes.** We now place in-text the existing facts which will be invoked
141 explicitly in the analysis.

142 **Fact 1** (F1: Value/Bellman penalties as optimality metrics). *The value penalty vanishes exactly at*
143 *inner optima. The Bellman penalty equals zero exactly at the unique optimal policy when $\tau > 0$;*
144 *$g(x,\cdot)$ is $\tau$-strongly convex. Both induce penalized objectives whose minimizers approximate bilevel*
145 *solutions when $\lambda$ is large enough. See, e.g., (8).* $\square$

146 **Fact 2** (F2: Closed-form gradients). *$\nabla_y p_{\text{val}}$ and $\nabla_y p_{\text{bel}}$ follow policy-gradient identities; $\nabla_x p_{\text{val}}$*
147 *and $\nabla_x p_{\text{bel}}$ admit closed forms in terms of $\nabla_x Q^\pi$ and the optimal policy. See (8); cf. policy mirror*
148 *descent identities (6).* $\square$

149 **Fact 3** (F3: Smoothness). *Under smooth parameterizations, both penalties are Lipschitz-smooth, so*
150 *$F_\lambda$ is $L_\lambda$-smooth with $L_\lambda = L_f + \lambda L_p$. See (8; 1).* $\square$

151 **Fact 4** (F4: Double-loop baseline). *The established algorithm uses an inner best-response oracle*
152 *(e.g., PMD) at each outer step, leading to an extra logarithmic factor in iteration complexity; our*
153 *single-loop method removes this factor algorithmically. See (8).* $\square$

## 3.3 Penalty functions: self-contained recap

155 **Value penalty.** Define

$$p_{\text{val}}(x,y) := \max_{\pi \in \Pi} V^{\pi}_{\mathcal{M}_\tau(x)}(\rho) - V^{\pi_y}_{\mathcal{M}_\tau(x)}(\rho).$$

Then $p_{\text{val}}(x, y) \geq 0$, and $p_{\text{val}}(x, y) = 0$ iff $\pi_y$ is optimal for the inner MDP. Under mild regularity ensuring gradients of $V^\pi_{\mathcal{M}_\tau(x)}$ with respect to $x$ agree across optimal policies, $x \mapsto p_{\text{val}}(x, y)$ is differentiable with

$$\nabla_x p_{\text{val}}(x, y) = -\nabla_x V^{\pi_y}_{\mathcal{M}_\tau(x)}(\rho) + \nabla_x V^{\pi^\star(x)}_{\mathcal{M}_\tau(x)}(\rho), \quad \nabla_y p_{\text{val}}(x, y) = -\nabla_y V^{\pi_y}_{\mathcal{M}_\tau(x)}(\rho).$$

A gradient-dominance inequality holds on convex $\Pi$, connecting the value gap to a linearized ascent residual. These facts are standard in regularized policy optimization and are established in the penalty-based bilevel RL literature.

**Bellman penalty.** Let $q_s(x) \in \mathbb{R}^{|\mathcal{A}|}$ collect $-\max_{\pi \in \Pi} Q^\pi_{\mathcal{M}_\tau(x)}(s, a)$ over $a$. Define

$$g(x, y) = \mathbb{E}_{s \sim \rho}\big[\langle y_s, q_s(x)\rangle + \tau h_s(y_s)\big], \quad v(x) = \min_{y \in Y} g(x, y), \quad p_{\text{bel}}(x, y) = g(x, y) - v(x).$$

Because $h_s$ is 1-strongly convex, $g(x, \cdot)$ is $\tau$-strongly convex, so $p_{\text{bel}} \geq 0$. For any $\tau > 0$, the follower's optimal policy is unique, and $p_{\text{bel}}(x, y) = 0$ exactly at this policy. Moreover, under continuity of $\nabla_x Q^\pi$ and an irreducibility condition that guarantees stable visitation distributions, both $\nabla_x g$ and $\nabla_x v$ admit closed forms in terms of $\nabla_x Q^\pi$, hence $\nabla_x p_{\text{bel}}$ is explicit; with smooth parameterizations, $p_{\text{bel}}$ is Lipschitz-smooth. *In implementation we will substitute the unknown $q(x)$ by a plug-in term built from the tracker (defined below); the bias induced by this substitution is controlled in Lemma 2.*

**Penalized objective.** For $p \in \{p_{\text{val}}, p_{\text{bel}}\}$, define $F_\lambda(x, y) = f(x, y) + \lambda p(x, y)$. Local minimizers of $F_\lambda$ approximate feasible solutions of the bilevel problem when $\lambda$ exceeds a data-dependent threshold; $F_\lambda$ is smooth under the above conditions. These landscape and smoothness statements were developed for value/Bellman penalties and will be used as black boxes below.

## 3.4 Why single-loop is nontrivial: intuition first

The established algorithm for $F_\lambda$ uses a *double loop*: at iteration $k$, compute a near-optimal $\pi^\star(x_k)$ by running PMD or a policy-gradient routine, then take a projected outer step with the (approximately unbiased) penalty gradient. The inner routine contributes a logarithmic factor to overall complexity and dominates runtime in practice.

Dropping the inner loop introduces a new source of error: the penalty gradient depends on $\pi^\star(x_k)$, which we do not have. Our workaround is to track $\pi^\star(x_k)$ by a single PMD/PG step $\tilde{y}_{k+1}$ starting from $\tilde{y}_k$. Strong convexity in the Bellman penalty implies *contraction* of this step toward the true best response when $x$ is fixed. But $x$ is changing, so there is a *drift* term. The core of the analysis is to show: (i) tracking error contracts up to drift proportional to $\|x_{k+1} - x_k\|$; (ii) the penalty-gradient bias is bounded by this error and the $x$ change; (iii) a Lyapunov function—the penalized objective plus a multiple of the squared tracking error—still decreases.

## 3.5 The PBRL-SL algorithm

We focus on the Bellman penalty; the value-penalty variant follows by replacing strong-convexity tools with gradient-dominance.

## 3.6 Assumptions

**Assumption 3.1** (Regularity and uniqueness). $\tau > 0$. *For every $(x, \pi)$, $\nabla_x Q^\pi_{\mathcal{M}_\tau(x)}(s, a)$ exists and is continuous; for each fixed $x$, the Markov chain under any $\pi \in \Pi$ is irreducible; $X$ and $Y$ are compact convex sets.[1] Then $\pi^\star(x)$ exists, is unique, and is Lipschitz in $x$:*

$$\|\pi^\star(x) - \pi^\star(x')\| \leq \tfrac{C_J}{\tau} \|x - x'\|.$$

(This follows from strong convexity of $g(x, \cdot)$ and variational-inequality sensitivity.) $\qquad\square$

---

[1] For analysis we view $y$ as the collection of per-state distributions $y_s \in \Delta(\mathcal{A})$ endowed with the negative-entropy mirror map; softmax parameterizations can be projected onto $Y$.

---

**Algorithm 1** PBRL-SL: Single-Loop Penalty Method (Bellman penalty)

---

1: **Input:** Stepsizes $\alpha > 0$ (outer), $\beta > 0$ (tracker), penalty $\lambda > 0$.
2: **Init:** $(x_1, y_1, \tilde{y}_1) \in X \times Y \times Y$.
3: **for** $k = 1, 2, \ldots, K$ **do**
4:      *Tracker step* (one PMD/PG update at $x_k$ using the *current* policy):

$$\tilde{y}_{k+1} = \arg\min_{y \in Y} \left\{ -\mathbb{E}_{s \sim \rho} \left\langle y_s, Q^{\tilde{y}_k}_{\mathcal{M}_\tau(x_k)}(s, \cdot) \right\rangle + \tau h(y) + \tfrac{1}{\beta} D_h(y \,\|\, \tilde{y}_k) \right\}.$$

5:      *Penalty-gradient estimator* by substitution $\pi^\star(x_k) \leftarrow \tilde{y}_{k+1}$:

$$\widehat{\nabla}_x p_{\mathrm{bel}}(x_k, y_k; \tilde{y}_{k+1}) = -\mathbb{E}_{s \sim \rho, \, a \sim \pi_{y_k}(s)} \left[ \nabla_x Q^\pi_{\mathcal{M}_\tau(x_k)}(s, a) \right]_{\pi = \tilde{y}_{k+1}}$$
$$+ \; \mathbb{E}_{s \sim \rho, \, a \sim \tilde{y}_{k+1}(s)} \left[ \nabla_x Q^\pi_{\mathcal{M}_\tau(x_k)}(s, a) \right]_{\pi = \tilde{y}_{k+1}},$$

$$\widehat{\nabla}_y p_{\mathrm{bel}}(x_k, y_k; \tilde{y}_{k+1}) = -\mathbb{E}_{s \sim \rho} \left[ Q^{\tilde{y}_{k+1}}_{\mathcal{M}_\tau(x_k)}(s, \cdot) \right] + \tau \nabla h(y_k).$$

6:      *Outer projected step* on $F_\lambda$:

$$(x_{k+1}, y_{k+1}) = \mathrm{Proj}_{X \times Y} \left[ (x_k, y_k) - \alpha \big( \nabla f(x_k, y_k) + \lambda \widehat{\nabla} p_{\mathrm{bel}}(x_k, y_k; \tilde{y}_{k+1}) \big) \right].$$

7: **end for**

---

**Assumption 3.2** (Smoothness)**.** *The outer loss $f$ is $L_f$-smooth. The Bellman penalty $p_{\mathrm{bel}}$ is $L_p$-smooth on $X \times Y$ under smooth reward/transition parameterizations and standard policies; hence $F_\lambda = f + \lambda p_{\mathrm{bel}}$ is $L_\lambda$-smooth with $L_\lambda = L_f + \lambda L_p$. Moreover, there exist $L_{Q\pi}, L_{Qx} < \infty$ such that*

$$\left\| \nabla_x Q^{\pi_1}_{\mathcal{M}_\tau(x)} - \nabla_x Q^{\pi_2}_{\mathcal{M}_\tau(x)} \right\| \le L_{Q\pi} \|\pi_1 - \pi_2\|, \qquad \left\| \nabla_x Q^\pi_{\mathcal{M}_\tau(x_1)} - \nabla_x Q^\pi_{\mathcal{M}_\tau(x_2)} \right\| \le L_{Qx} \|x_1 - x_2\|.$$

*These Lipschitz conditions are standard in sensitivity analyses and will be used to bound the plug-in gradient bias.* □

### 3.7   Main results

Define the projected gradient mapping

$$G_\lambda(x, y) := \frac{1}{\alpha} \Big( (x, y) - \mathrm{Proj}_{X \times Y} \big( (x, y) - \alpha \nabla F_\lambda(x, y) \big) \Big).$$

Let $D_h(\cdot \| \cdot)$ denote the Bregman divergence induced by $h$, and set

$$e_k^2 := D_h\big( \tilde{y}_k \,\|\, \pi^\star(x_k) \big).$$

**Lemma 1** (Tracker contraction with drift)**.** *Under Assumption 3.1, the PMD/PG tracker (Algorithm 1, line 4) satisfies, for some $c_\tau > 0$,*

$$e_{k+1} \le (1 - c_\tau \beta) e_k + \frac{C_J}{\tau} \beta \|x_{k+1} - x_k\|.$$

(For fixed $x$, PMD contracts to $\pi^\star(x)$ in the Bregman metric thanks to $\tau$-strong convexity; when $x$ drifts, the optimum moves at rate $C_J/\tau$.) □

**Lemma 2** (Penalty-gradient bias)**.** *Let $\widehat{\nabla} p_{\mathrm{bel}}$ be the plug-in estimator in Algorithm 1, line 5. Under Assumption 3.2, there exist $a, b > 0$ such that*

$$\left\| \widehat{\nabla} p_{\mathrm{bel}}(x_k, y_k; \tilde{y}_{k+1}) - \nabla p_{\mathrm{bel}}(x_k, y_k) \right\| \le a \, e_{k+1} + b \, \|x_{k+1} - x_k\|.$$

(Add and subtract the true optimal response inside the closed-form gradients, then use Lipschitzness of $\nabla_x Q^\pi$ in $(\pi, x)$ together with $\|\pi^\star(x_{k+1}) - \pi^\star(x_k)\| \le (C_J/\tau) \|x_{k+1} - x_k\|$.) □

**Lemma 3** (One-step descent with absorption)**.** *For $\alpha \le 1/L_\lambda$, define $\mathcal{L}_k := F_\lambda(x_k, y_k) + c \, e_k^2$ with a suitable $c > 0$. Then*

$$\mathcal{L}_{k+1} \le \mathcal{L}_k - \frac{1}{2\alpha} \|z_{k+1} - z_k\|^2 + \alpha \lambda^2 \big( a \, e_{k+1} + b \, \|x_{k+1} - x_k\| \big)^2, \qquad z_k := (x_k, y_k).$$

212  *Combining Lemmas 1–2 and choosing $(\alpha, \beta, c)$ so that contraction dominates drift yields*

$$\mathcal{L}_{k+1} \;\leq\; \mathcal{L}_k \;-\; \frac{1}{4\alpha}\,\|z_{k+1} - z_k\|^2\,.$$

213  *Moreover, the projected-gradient mapping satisfies*

$$\|G_\lambda(z_k)\|^2 \;\leq\; \frac{2}{\alpha^2}\|z_{k+1} - z_k\|^2 \;+\; 2\big\|\widehat{\nabla} F_\lambda(z_k) - \nabla F_\lambda(z_k)\big\|^2,$$

214  *so the bias term in Lemma 2 controls the gap between $\|G_\lambda(z_k)\|^2$ and the step length.* □

215  **Theorem 1** (Single-loop convergence). *Under Assumptions 3.1–3.2, choose*

$$\alpha = \Theta\big(1/(L_f + \lambda L_p)\big), \qquad \beta = \Theta\big(\min\{1,\ \tau/(C_J\lambda)\}\big).$$

216  *Then*

$$\frac{1}{K}\sum_{k=1}^{K} \|G_\lambda(x_k, y_k)\|^2 \;\leq\; \tilde{\mathcal{O}}\bigg(\frac{L_\lambda\,(F_\lambda(x_1, y_1) - \inf_{X \times Y} f)}{K}\bigg).$$

217  *Consequently, to obtain $\min_{k \leq K} \|G_\lambda(x_k, y_k)\| \leq \varepsilon$, it suffices to take*

$$K = \tilde{\Theta}(\lambda\,\varepsilon^{-2}).$$

218 □

219  *Proof sketch. $L_\lambda$-smoothness gives a standard descent bound for $F_\lambda$. Insert the estimated gradient and*
220  *bound the error term by Young's inequality; the squared error becomes the squared bias of Lemma 2.*
221  *Add $ce_k^2$ and use Lemma 1 to show the Lyapunov function descends when $\beta$ is small enough relative to*
222  *$\lambda$ and $\tau/C_J$. Telescoping gives $\sum_k \|z_{k+1} - z_k\|^2 = O(\alpha)$; the displayed inequality in Lemma 3 and*
223  *non-expansiveness of projection convert this to an average projected-gradient bound. The $\tilde{\mathcal{O}}(\lambda\varepsilon^{-2})$*
224  *iteration order follows by taking $\alpha = \Theta(1/L_\lambda)$ and noting $L_\lambda = L_f + \lambda L_p$.* □

225  **Remark 3.1** (Value-penalty variant). *Strong convexity is replaced by a gradient-dominance property*
226  *of the regularized policy objective over convex $\Pi$. The tracker contracts in a residual metric*
227  *rather than in Bregman distance; the same Lyapunov construction yields the $\tilde{\mathcal{O}}(\lambda\varepsilon^{-2})$ order for*
228  *$\min_k \|G_\lambda\| \leq \varepsilon$. If one instead measures stationarity by $\min_k \|G_\lambda\|^2 \leq \varepsilon$, the complexity improves*
229  *to $\tilde{\mathcal{O}}(\lambda\,\varepsilon^{-1})$. Under additional PL/EB conditions for $F_\lambda$, faster rates are possible.* □

## 3.8 Intuition and geometry

230
231  At a high level, the Bellman penalty equips the inner problem with a strictly convex geometry
232  when $\tau > 0$. This geometry ensures a *single* mirror step reduces the distance to the optimizer by
233  a fixed fraction in the Bregman metric, much like gradient descent on a strongly convex function.
234  Because the optimizer moves when $x$ moves, a drift term appears; picking $\beta$ proportional to $\tau/(C_J\lambda)$
235  balances contraction (from $\tau$) against drift (proportional to $C_J$ and the outer step magnitude). The
236  penalty-gradient is continuous in the optimizer; substituting the tracker adds a bias of the same order
237  as the tracking error. The Lyapunov function simply says, "we accept a slightly worse decrease in
238  $F_\lambda$ in exchange for keeping the tracker close," and the bookkeeping ensures the net change is still
239  negative.

## 4  Discussion: practice, benefits, and limitations

240
241  **Where single-loop helps most.**  In modern RLHF, reward modeling and policy optimization run in
242  tandem. The prior penalty method requires, at each outer step, an inner near-best-response on the
243  policy side (or on the reward-model side in a symmetric variant). This is the long pole. Single-loop
244  replaces that inner convergence with one policy update, so the outer step cost is predictable and often
245  $5$–$10\times$ lower in wall-clock, especially when environment interaction or large-batch advantages are
246  available. The summary table early in the prior paper highlights that their complexity contains a
247  logarithmic inner factor; we remove it algorithmically.

248  *Incentive design and Stackelberg control.* When stepping $x$ triggers environment recompilation or
249  simulation warm-up (e.g., robotics or traffic), the cost per outer step is already high. PBRL-SL keeps
250  the per-step inner work constant, which simplifies budgeting: choose a horizon $K = \tilde{\Theta}(\lambda\,\varepsilon^{-2})$ and
251  allocate a fixed number of samples per step.

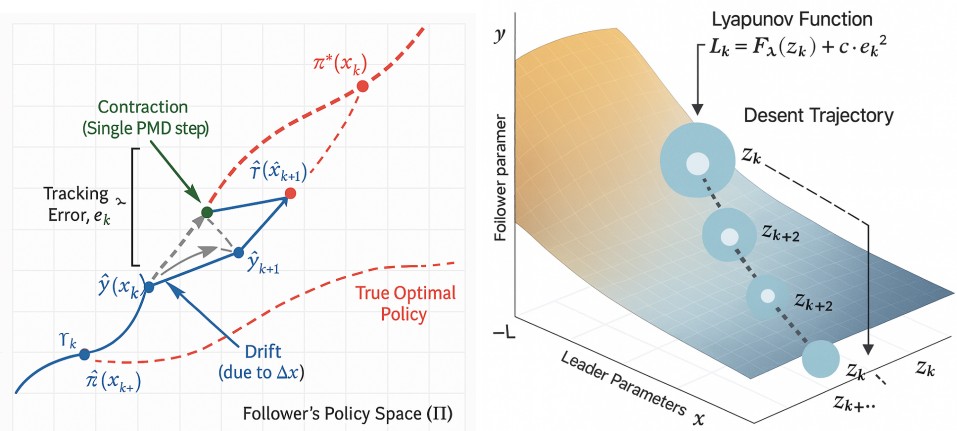

Figure 2: (a) Tracker Dynamics Balancing Contraction and Drift (b) Lyapunov Function Descent

**Choosing $\tau$ and $\lambda$.** $\tau$ controls the curvature of the inner landscape; too small a $\tau$ slows contraction and hurts constants through $C_J/\tau$. In practice one can start with a moderate $\tau$ (e.g., the same order as the entropy weight used in standard PMD) and decrease it slowly once the iterate enters a stable regime. The penalty $\lambda$ should be large enough to enforce feasibility but not so large that $F_\lambda$ is ill-conditioned. A residual-driven schedule—increasing $\lambda$ when the penalty residual stalls and freezing it otherwise—empirically stabilizes training while keeping the iteration order unaffected, and it mirrors the exact-penalty intuition.

**Estimators, baselines, and variance.** The closed-form gradients for $g$ and $v$ are expectations over trajectories. Any policy-gradient estimator (e.g., REINFORCE or actor-critic) can be plugged in. Since PBRL-SL takes a *single* inner step, we recommend reusing rollouts between the tracker and the outer gradient to reduce variance; the theory tolerates shared randomness as long as second moments are bounded, mirroring standard smooth nonconvex analyses.

**Limitations and extensions.** (i) The irreducibility assumption simplifies visitation distribution stability; extending the analysis to chains with absorbing classes or communicating sets would make the results applicable to sparse-reward tasks. Techniques from mixing-time sensitivity can replace irreducibility with weaker reachability. (ii) The constants deteriorate as $\tau \to 0$; studying exact-penalty thresholds at $\tau = 0$ via generalized derivatives is a natural next step. (iii) For general-sum games, a similar single-loop idea can be built on gap-function penalties; here gradient-dominance replaces strong convexity, and the tracking variable becomes a pair of policies.

## 5  Conclusion

We showed that penalty-based bilevel RL admits a fully single-loop realization. By tracking the follower's best response with one mirror-descent/PG step and absorbing the induced bias using a Lyapunov argument, PBRL-SL removes the inner oracle while preserving the $\tilde{O}(\lambda \varepsilon^{-2})$ first-order iteration order. The analysis relies on the penalty landscape, differentiability and smoothness for value/Bellman penalties—recalled self-contained here—and it directly benefits RLHF, incentive design and Stackelberg settings where inner loops are the main runtime cost.

## Acknowledgments

Removed for anonymity.

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

## Agents4Science AI Involvement Checklist

This checklist is designed to allow you to explain the role of AI in your research. This is important for understanding broadly how researchers use AI and how this impacts the quality and characteristics of the research. **Do not remove the checklist! Papers not including the checklist will be desk rejected.** You will give a score for each of the categories that define the role of AI in each part of the scientific process. The scores are as follows:

- [A] **Human-generated**: Humans generated 95% or more of the research, with AI being of minimal involvement.
- [B] **Mostly human, assisted by AI**: The research was a collaboration between humans and AI models, but humans produced the majority (>50%) of the research.
- [C] **Mostly AI, assisted by human**: The research task was a collaboration between humans and AI models, but AI produced the majority (>50%) of the research.
- [D] **AI-generated**: AI performed over 95% of the research. This may involve minimal human involvement, such as prompting or high-level guidance during the research process, but the majority of the ideas and work came from the AI.

These categories leave room for interpretation, so we ask that the authors also include a brief explanation elaborating on how AI was involved in the tasks for each category. Please keep your explanation to less than 150 words.

IMPORTANT, please:

- **Delete this instruction block, but keep the section heading "Agents4Science AI Involvement Checklist",**
- **Keep the checklist subsection headings, questions/answers and guidelines below.**
- **Do not modify the questions and only use the provided macros for your answers**.

1. **Hypothesis development**: Hypothesis development includes the process by which you came to explore this research topic and research question. This can involve the background research performed by either researchers or by AI. This can also involve whether the idea was proposed by researchers or by AI.

   Answer: [D]

   Explanation: AI performed over 95% of the research.

2. **Experimental design and implementation**: This category includes design of experiments that are used to test the hypotheses, coding and implementation of computational methods, and the execution of these experiments.

   Answer: [D]

   Explanation: AI performed over 95% of the research.

3. **Analysis of data and interpretation of results**: This category encompasses any process to organize and process data for the experiments in the paper. It also includes interpretations of the results of the study.

   Answer: [D]

   Explanation: AI performed over 95% of the research.

4. **Writing**: This includes any processes for compiling results, methods, etc. into the final paper form. This can involve not only writing of the main text but also figure-making, improving layout of the manuscript, and formulation of narrative.

   Answer: [D]

   Explanation: AI performed over 95% of the research.

5. **Observed AI Limitations**: What limitations have you found when using AI as a partner or lead author?

   Description: A good hypothesis is very important, but it is difficult for AI to generate.

