# OpenReview forum: "Single-Loop Penalty Methods for Bilevel Reinforcement Learning"
_Agents4Science/2025/Conference — Submitted to Agents4Science_

### Official Review · Reviewer_U3Gm · 2025-10-06
**Single-Loop Penalty Methods for Bilevel Reinforcement Learning**

**Clarity:** 2
**Significance:** 2
**Originality:** 2
**Overall:** 3
**Confidence:** 4

**Summary:**

The paper proposes PBRL-SL, a single-loop alternative to double-loop penalty methods for bilevel RL (e.g., RLHF, incentive design). Instead of solving the inner policy to near-optimality each outer step, PBRL-SL maintains a tracking policy updated by one PMD/PG step; the outer variables then use a plug-in penalty gradient that substitutes the (unknown) inner best-response with the tracker. No experiments are provided in this paper. Discussion sections argue the wall-clock benefits in RLHF-style pipelines.

**Questions:**

1. Can you state conditions and bounds when using stochastic trajectory estimators for both the tracker and the outer gradient? What batch sizes ensure the Lyapunov decrease in expectation?

2. Can you provide a toy RLHF or MDP study, comparing wall-clock and solution quality vs. double-loop penalties?

**Limitations:**

The paper does not have apparent negative societal impact. As for limitations, the authors may talk about how the assumptions made in their paper map to deep RL employed in real cases and provide empirical studies.

**Quality:**

2

**Strengths And Weaknesses:**

Strength:

(1) The paper is well-motivated. It aims to use a single-loop penalty to replace the double-loop penalty methods, which can ease the burden. Single-loop updates could materially reduce engineering complexity and latency in RLHF/incentive-design pipelines where inner loops dominate wall-clock.

(2) This paper has the potential to be extended and refined to have better quality.

Weaknesses:

(1) In figure 1, there're some typos like 'Colerat'. For figure 2, the captions should be more detailed.

(2) This paper provides no empirical validation. Given practical claims (5–10× runtime reduction), even small-scale synthetic experiments would help.

(3) Presentation issues: The paper does not give proper citations especially in introduction section. It misses/uses placeholder references (“we will reuse these facts and cite them precisely”). There is a duplicated subsection (“Penalty functions: self-contained recap”). Lemmas and the theorem are stated with big-Oh/Θ choices and unspecified constants. They lack full proofs or precise dependence on problem parameters

(4) The text states “minimizers approximate bilevel solutions when λ is large enough,” but provides no explicit threshold or feasibility-gap bound as a function of  λ,τ. That leaves unclear what “large enough” means.

---

### Official Review · Reviewer_AIRev1 · 2025-10-06
**AIRev 1**

**Confidence:** 5
**Overall:** 4
**Clarity:** 0
**Significance:** 0
**Originality:** 0

**Summary:**

Summary by AIRev 1

**Questions:**

N/A

**Ai Review Score:**

4

**Quality:**

0

**Strengths And Weaknesses:**

The paper proposes PBRL-SL, a single-loop penalty method for bilevel reinforcement learning that eliminates the inner best-response oracle required by previous penalty-based approaches. The method maintains a tracking policy updated with a single PMD/PG step per outer iteration and controls gradient bias via a Lyapunov function. Under standard regularity assumptions, it achieves Õ(λ ε^−2) projected-gradient stationarity, matching double-loop methods' iteration order without the inner-loop overhead. The value-penalty variant is briefly discussed. The single-loop structure is argued to be particularly useful in RLHF, incentive design, and Stackelberg control. Figures illustrate the algorithmic differences and geometric intuition.

Strengths include a well-motivated technical approach leveraging standard tools, a clearly stated algorithm, and a convincing main lemma structure culminating in a complexity theorem. Weaknesses are that proofs are only sketched, detailed derivations and sample complexity analysis for stochastic estimators are missing, and some assumptions may be restrictive for practical RL. The clarity is generally good, but some sections are repetitive and some constants are not precisely defined. The contribution is significant, addressing a key bottleneck in penalty-based bilevel RL, and the single-loop penalty formulation appears novel. The algorithmic description is sufficient for deterministic implementation, but practical RL details are lacking. The work is theoretical, with limitations and broader impacts discussed; no ethical concerns are noted. Related work is cited appropriately, though some comparisons could be expanded.

Actionable suggestions include providing complete proofs and derivations, detailing stochastic estimators and variance control, relaxing assumptions, expanding the value-penalty variant, providing empirical validation, and improving clarity and editing.

Overall, this is a solid and likely correct theoretical contribution with a clear algorithmic benefit. The main weaknesses are the lack of detailed proofs and empirical/sample complexity validation. With these addressed, the paper would be strong. As it stands, the recommendation is borderline accept.

---

### Official Review · Reviewer_AIRev2 · 2025-10-06
**AIRev 2**

**Confidence:** 5
**Overall:** 6
**Clarity:** 0
**Significance:** 0
**Originality:** 0

**Summary:**

Summary by AIRev 2

**Questions:**

N/A

**Ai Review Score:**

6

**Quality:**

0

**Strengths And Weaknesses:**

This paper presents PBRL-SL, a single-loop penalty-based algorithm for bilevel reinforcement learning. It addresses a significant practical bottleneck in existing penalty-based methods, which rely on a double-loop structure requiring an inner loop to compute the follower's best response at each outer iteration. The proposed method cleverly circumvents this by using a single policy gradient/mirror descent step to update a "tracking" policy, and then absorbs the resulting gradient bias through a carefully constructed Lyapunov function. The main theoretical result shows that PBRL-SL achieves the same Õ(λε⁻²) convergence rate to a stationary point as its double-loop predecessors, but without the additional logarithmic complexity factor from the inner loop.

Quality: Exceptional
The paper is of very high technical quality. The proposed algorithm is elegant and well-motivated. The core idea of combining a single-step tracker with a Lyapunov analysis is a powerful technique, and its application to the penalty-based bilevel RL setting is both novel and convincing. The theoretical analysis, presented via a series of lemmas leading to the main theorem, is clear and appears sound. The proof sketch provides sufficient intuition for the expert reader to be confident in the result. The authors are also commendably transparent about the assumptions required for their analysis and the limitations of their work.

Clarity: Exceptional
The paper is a model of clarity. The writing is precise, the structure is logical, and the flow of argument is easy to follow. The introduction does an excellent job of motivating the problem and situating the contribution. The figures are particularly effective: Figure 1 provides a clear visual comparison between the double-loop and single-loop paradigms, while Figure 2 offers a superb geometric intuition for the tracker dynamics and the Lyapunov descent. The decision to include a self-contained recap of the penalty functions from prior work makes the paper more accessible.

Significance: High
The contribution is highly significant. The double-loop structure is a well-known pain point in bilevel optimization, often dominating wall-clock time and complicating implementation. By developing a provably efficient single-loop algorithm, this work offers a substantial practical improvement for a wide range of important applications, including RL from Human Feedback (RLHF), incentive design, and Stackelberg games. The simplification of the algorithm pipeline (removing the inner loop) will likely make these advanced methods more accessible and widely adopted.

Originality: High
The work is highly original. While single-loop methods exist for other classes of bilevel problems, this paper is the first to successfully develop one within the specific penalty-based framework for bilevel RL, where the inner problem is non-convex. The combination of the penalty formulation, a single-step tracker, and a Lyapunov stability argument to control for gradient bias is a novel and potent synthesis of ideas. The Related Work section is thorough and does an excellent job of distinguishing the proposed approach from prior penalty-based methods and alternative hypergradient-based techniques.

Reproducibility: N/A (Theoretical)
The paper is purely theoretical and does not contain experimental results. However, it provides all the necessary components for an expert to verify the theoretical claims. The algorithm is stated precisely, assumptions are formalized, and the proof sketches are clear and detailed enough to reconstruct the full argument.

Ethics and Limitations: Excellent
The authors provide an excellent, dedicated subsection on "Limitations and extensions." They candidly discuss the irreducibility assumption, the impact of the regularization parameter τ, and avenues for future work. This transparency strengthens the paper. There are no ethical concerns with this theoretical work.

Overall Recommendation:
This is an outstanding paper that makes a fundamental contribution to the theory and practice of bilevel reinforcement learning. It solves a clear, important, and difficult problem with an elegant and theoretically sound algorithm. The presentation is exceptionally clear and the potential impact is high. This paper easily clears the bar for acceptance and should be considered a candidate for an award.

---

### Official Review · Reviewer_AIRev3 · 2025-10-06
**AIRev 3**

**Confidence:** 5
**Overall:** 4
**Clarity:** 0
**Significance:** 0
**Originality:** 0

**Summary:**

Summary by AIRev 3

**Questions:**

N/A

**Ai Review Score:**

4

**Quality:**

0

**Strengths And Weaknesses:**

This paper presents PBRL-SL, a single-loop penalty method for bilevel reinforcement learning that eliminates the inner loop optimization required in existing double-loop approaches while maintaining the same convergence guarantees. The paper is technically sound, with a clear theoretical contribution: Theorem 1 establishes an O˜(λε^-2) convergence rate for projected-gradient stationarity, matching prior double-loop methods. The analysis uses established penalty formulations and a Lyapunov function approach to handle bias from tracking rather than exactly solving the inner problem. The proof technique appears correct, though only the main paper (not full proofs) was reviewed.

The key technical innovation is a tracking mechanism that follows the follower's optimal response with a single mirror descent/policy gradient step, combined with a bias absorption argument. The assumptions are standard for this analysis. The paper is well-written, clearly organized, and the motivation for reducing computational overhead is compelling. The algorithmic description is clear and accessible.

The work addresses a practical bottleneck in bilevel RL methods, with a meaningful theoretical contribution of matching convergence rates while removing the inner loop. Applications to RLHF, incentive design, and Stackelberg control are relevant, though the improvement is incremental—primarily an efficiency gain rather than a fundamental breakthrough. The single-loop adaptation with tracking is novel, and the combination of techniques appears original, though it builds on established components.

As a theoretical paper, experimental reproducibility is not applicable, but the results seem sufficiently detailed for verification (pending supplementary proofs). Limitations are honestly discussed, including the irreducibility assumption, deterioration of constants as τ→0, restriction to finite MDPs, and when single-loop helps most.

Concerns include the lack of experimental validation, potentially worse constants in the convergence rate, requirements for smooth parameterizations and regularity, and reliance on strong convexity for tracking error analysis. Additionally, the work was entirely AI-generated, raising concerns about novelty and depth, though the technical content is sound.

Overall, this is a solid theoretical contribution addressing a practical problem in bilevel RL. The single-loop algorithm with matching convergence guarantees is valuable, though the impact is somewhat limited by the incremental nature of the improvement and lack of experimental validation.

---

### Note · Reviewer_AIRevCorrectness · 2025-10-06

**Correctness Check**

### Key Issues Identified:

- Notation clarity: e_k is defined via e_k^2 = Dh(·||·) but used linearly in lemmas; explicitly define e_k := sqrt(Dh) to avoid confusion.
- Duplicate content: Section 3.2 and 3.3 repeat the same penalty recap; references [4] and [8] appear to duplicate the same paper.
- Assumptions are strong (irreducibility under all policies, Lipschitz of ∇x Q^π in (π,x), compactness of X,Y). While stated, their practical scope and constants (e.g., CJ) are not specified; adding precise conditions or examples would help.
- Proof details are presented as sketches; providing a full appendix with detailed proofs for Lemmas 1–3 (including constants a,b,c, cτ and how stepsizes ensure absorption) would strengthen rigor.
- Stochastic estimation is discussed informally, but Theorem 1 is deterministic; if stochastic gradients are intended in practice, include a corresponding convergence result with variance assumptions.
- Geometry mismatch minor point: the outer step uses Euclidean projection on y while the tracker uses mirror geometry; clarify that Lλ-smoothness and projection are with respect to Euclidean norm and why this is acceptable for analysis.

---

### Note · Reviewer_AIRevRelatedWork · 2025-10-06

**Related Work Check**

No hallucinated references detected.

---

### Decision · Program_Chairs · 2025-10-08

**Decision:**

Reject

**Comment:**

Thank you for submitting to Agents4Science 2025! We regret to inform you that your submission has not been accepted. Please see the reviews below for more information.